# Structural Basis for the Enhanced Infectivity and Immune Evasion of Omicron Subvariants

**DOI:** 10.3390/v15061398

**Published:** 2023-06-20

**Authors:** Yaning Li, Yaping Shen, Yuanyuan Zhang, Renhong Yan

**Affiliations:** 1Center for Infectious Disease Research, Westlake Laboratory of Life Sciences and Biomedicine, Key Laboratory of Structural Biology of Zhejiang Province, School of Life Sciences, Westlake University, Hangzhou 310024, China; liyaning0405@163.com (Y.L.); shenyaping@westlake.edu.cn (Y.S.); zhangyuanyuan36@westlake.edu.cn (Y.Z.); 2Beijing Advanced Innovation Center for Structural Biology, Tsinghua-Peking Joint Center for Life Sciences, School of Life Sciences, Tsinghua University, Beijing 100084, China; 3Key University Laboratory of Metabolism and Health of Guangdong, Department of Biochemistry, School of Medicine, Southern University of Science and Technology, Shenzhen 518055, China

**Keywords:** SARS-CoV-2, Omicron subvariant, spike, ACE2, immune evasion

## Abstract

The Omicron variants of SARS-CoV-2 have emerged as the dominant strains worldwide, causing the COVID-19 pandemic. Each Omicron subvariant contains at least 30 mutations on the spike protein (S protein) compared to the original wild-type (WT) strain. Here we report the cryo-EM structures of the trimeric S proteins from the BA.1, BA.2, BA.3, and BA.4/BA.5 subvariants, with BA.4 and BA.5 sharing the same S protein mutations, each in complex with the surface receptor ACE2. All three receptor-binding domains of the S protein from BA.2 and BA.4/BA.5 are “up”, while the BA.1 S protein has two “up” and one “down”. The BA.3 S protein displays increased heterogeneity, with the majority in the all “up” RBD state. The different conformations preferences of the S protein are consistent with their varied transmissibility. By analyzing the position of the glycan modification on Asn343, which is located at the S309 epitopes, we have uncovered the underlying immune evasion mechanism of the Omicron subvariants. Our findings provide a molecular basis of high infectivity and immune evasion of Omicron subvariants, thereby offering insights into potential therapeutic interventions against SARS-CoV-2 variants.

## 1. Introduction 

The emergence of SARS-CoV-2 variants, particularly the Omicron variant of concern (VOC), continues to pose a significant health threat due to their increased transmissibility and ability to evade the immune response [1,2,3]. The Omicron variant has developed into multiple sublineages, including the original BA.1 (B.1.1.529.1), BA.2 (B.1.1.529.2), BA.3 (B.1.1.529.3), BA.4, and BA.5. The BA.1 sublineage has rapidly outcompeted the Delta variant across the world since the end of 2021 [4]. Approximately two months later, the BA.2 subvariant, known for its heightened transmissibility, replaced BA.1 and spread worldwide [1]. Then, the BA.4/BA.5 sublineages, which share the same S protein mutations (hereafter referred to as BA.5), were identified in Botswana and South Africa, demonstrating even greater transmission advantages compared to the BA.2 subvariant [5]. Additionally, the BA.3 sublineage is at a low prevalence for the time being but continues to spread [6].

The five sublineages of Omicron exhibit numerous mutations in the S protein, which is responsible for receptor recognition and facilitates fusion with host cells [7,8]. BA.1, BA.2, BA.3, and BA.4/BA.5 sublineages contain 37, 31, 33, and 34 mutations, respectively, in the S protein compared to the original strain (hereafter referred to as the WT strain) (Figure 1A). These mutations have the potential to enhance the variants’ transmission capability [7]. Among these, 15 mutations in BA.1 are located on the receptor-binding domain (RBD), which is responsible for direct association with angiotensin-converting enzyme 2 (ACE2), the surface receptor on the host cells [9,10]. In addition to 12 common mutations with BA.1, BA.2 has 4 distinct ones—S371F, T376A, D405N, and R408S—on the RBD. BA.5 might evolve from BA.2 and has three additional mutations—L452R, F486V, and R493Q—on the RBD. Additionally, BA.3 has combined mutations from BA.1 and BA.2 (Figure 1A). Recently, a newly emerged sublineage, Omicron XBB, characterized by high antibody evasion abilities, has gained dominance in the population [11].

Despite gaining a large number of mutations, all these Omicron subvariants still exploit ACE2 as the host receptor [12,13,14]. For SARS-CoV-2 infection, the S protein is cleaved into the S1 and S2 subunits by the host furin protease [15,16,17,18,19]. The S1 subunit contains the N-terminal domain (NTD) and RBD. The RBD may exhibit “up” and “down” conformations in reference to the viral surface, and only the “up” RBD can expose the binding site which is responsible for mediating the interaction with the peptidase domain (PD) of ACE2 [9]. This interaction triggers the conformational changes of the trimeric S protein to further expose the fusion peptide in S2, which then facilitates membrane fusion with host cells [9,20,21,22].

Omicron sublineages show increased immune escape capability [23]. BA.1 was reported to evade neutralization induced by various vaccines or infection with other SARS-CoV-2 variants [24,25,26]. Some patients who recovered from BA.1 were then infected by BA.2 within a short period [27,28], suggesting that the antibodies generated from the early Omicron BA.1 infection might fail to neutralize BA.2. Additionally, BA.4/5 exhibits compromised neutralization to the sera from triple AstraZeneca- or Pfizer-vaccinated individuals than BA.1 and BA.2 subvariants [29]. Indeed, the BA.2 and BA.5 subvariant can escape from several neutralizing antibodies, including S309 (sotrovimab, short as S309) that can effectively neutralize Alpha, Beta, Gamma, Delta, and BA.1 of SARS-CoV-2 [12,23]. S2K146 could mediate broad sarbecovirus neutralization including BA.2 and BA.3, while largely compromising the neutralization to BA.5 [30,31]. Fortunately, the recently authorized LY-CoV1404 (bebtelovimab) remains potent in neutralizing all Omicron sublineages [23,31,32].

The mutations on the S protein of different Omicron subvariants likely contribute to their modified characteristics in terms of receptor recognition, transmission, and immune escape. Consequently, it is crucial to identify the specific mutations on the S protein that account for these alterations. While structures of the BA.1 S protein in complex with ACE2 have been reported [13,33,34], additional structures involving BA.2, BA.3, and BA.5 in complex with ACE2 are necessary to establish the molecular foundation for the heightened transmissibility and immune evasion observed in Omicron mutations [6]. 

To address this crucial question, our study aimed to systematically examine structures of ACE2-bound S protein from BA.1, BA.2, BA.3, and BA.5. We show that all three RBDs of the trimeric S protein from BA.2 and BA.5 are “up”, while BA.1 has two “up” and one “down”. Although the majority of the BA.3 S proteins have three “up” RBDs, a small portion has two “up” and one “down”. This analysis provides an immediate explanation for their varying infectivities, as the “up” RBDs are to be recognized by ACE2. We also show that the shift of glycosylation moieties may lead to weakened neutralization of S309, and two mutations, L452R and F486V, of BA.5 decrease the binding to S2K146.

## 2. Materials and Methods

### 2.1. Protein Expression and Purification

The extracellular domains (ECDs) (1–1208 amino acid) of the S protein of SARS-CoV-2 Omicron variants BA.1/2/3/5 were cloned into the pCAG vector (Invitrogen, Waltham, MA, USA) with six proline substitutions at residues 817, 892, 899, 942, 986, and 987 and a C-terminal T4 fibritin trimerization motif followed by 10 × His tag, respectively. A “GSAS” mutation at residues 682 to 685 was introduced into ECD to prevent the host furin protease digestion. These constructs were hereafter referred to as BA.1/2/3/5-S [35]. 

The peptidase domain (PD) (19–615 amino acid) of human ACE2 was also cloned into the pCAG vector (Invitrogen) with an N-terminal signal peptide of secreted luciferase and a C-terminal Flag tag. The mutants were generated with a standard two-step PCR-based strategy [36]. All the plasmids used to transfect cells were prepared by GoldHi EndoFree Plasmid Maxi Kit (CWBIO) [37]. 

The recombinant protein was overexpressed using the HEK293F mammalian cells at 37 °C under 5% CO_2_ in a Multitron-Pro shaker (Infors, Bottmingen, Switzerland, 130 rpm). When the cell density reached 2.0 × 10^6^ cells/mL, the plasmid was transiently transfected into the cells. To transfect one liter of cell culture, about 1.5 mg of the plasmid was premixed with 3 mg of polyethylenimines (PEIs) (Polysciences) in 50 mL of fresh medium for 15 min before adding to cell culture. Sixty hours after transfection, cells were removed and medium was collected by centrifugation at 4000× *g* for 15 min. 

The secreted ECD of the S protein were purified by Ni-NTA affinity resin (Qiagen, Hilden, Germany). The nickel resin loaded was rinsed with wash buffer 1 containing 25 mM HEPES (pH 7.0) and 500 mM NaCl and washed with wash buffer 2 containing 25 mM HEPES (pH 7.0), 150 mM NaCl, and 30 mM imidazole. Protein was eluted by wash buffer 2 plus 270 mM imidazole. Then the Ni-NTA eluent of ECD was subjected to size-exclusion chromatography (Superose 6 Increase 10/300 GL, GE Healthcare, Chicago, IL, USA) in buffer containing 25 mM HEPES (pH 7.0), and 150 mM NaCl. The peak fractions were collected and stored at −80 °C. 

The secreted PD was purified by anti-FLAG M2 affinity resin (Sigma Aldrich, Saint-Quentin-Fallavier, France). After loading two times, the anti-FLAG M2 resin was washed with wash buffer 3 containing 25 mM HEPES (pH 7.0) and 150 mM NaCl. The protein was eluted with the wash buffer 3 plus 0.2 mg/mL flag peptide. The eluent of PD was then concentrated and subjected to size-exclusion chromatography (Superdex 200 Increase 10/300 GL, GE Healthcare) in buffer containing 25 mM HEPES (pH 7.0) and 150 mM NaCl.

The BA.1/2/3/5-S was incubated with PD at a molar ratio of about 1:6 for one hour. To remove excessive PD, the mixture was subjected to size-exclusion chromatography (Superose 6 Increase 10/300 GL, GE Healthcare) in buffer containing 25 mM HEPES (pH 7.0) and 150 mM NaCl. The peak fractions containing protein complex were collected for EM analysis.

### 2.2. Cryo-EM Sample Preparation and Data Acquisition

The BA. 2/3/5-SA were concentrated to ~1.5 mg/mL and applied to the grids. Aliquots (3.3 μL) of the protein were placed on glow-discharged holey carbon grids (Quantifoil Au R1.2/1.3). The grids were blotted for 3.0 s or 3.5 s and flash-frozen in liquid ethane cooled by liquid nitrogen with Vitrobot (Mark IV, Thermo Fisher Scientific, Waltham, MA, USA). The prepared grids were transferred to a Titan Krios operating at 300 kV equipped with Gatan K3 detector and GIF Quantum energy filter. Movie stacks were automatically collected using AutoEMation [38], with a slit width of 20 eV on the energy filter and a defocus range from −1.4 µm to −1.8 µm in super-resolution mode at a nominal magnification of 81,000×. Each stack was exposed for 2.56 s with an exposure time of 0.08 s per frame, resulting in a total of 32 frames per stack. The total dose rate was approximately 50 e^−^/Å^2^ for each stack. The stacks were motion corrected with MotionCor2 [39] and binned 2-fold, resulting in a pixel size of 1.087 Å/pixel. Meanwhile, dose weighting was performed [40]. The defocus values were estimated with Gctf [41].

### 2.3. Data Processing

The Cryo-EM structure of the S protein from BA.1-SA has been solved first [42], and the identical protocol was applied to the complexes of BA.2, BA.3 and BA.5. Particles for S-ECD bound with PD of ACE2 were automatically picked using Relion 3.0.6 [43,44,45,46] from manually selected micrographs. After 2D classification with Relion, good particles were selected and subject to multiple cycles of heterogeneous refinement without symmetry using cryoSPARC [47]. The good particles were selected and subjected to Local CTF Refinement with C1 symmetry, non-uniform refinement, resulting in the 3D reconstruction for the whole structures. For interface between the RBDs of S-ECD and PD, the dataset was subject to focused refinement with an adapted mask and reference on RBD-PD sub-complex to improve the map quality. Then the dataset of three RBD-PD sub-complexes were combined and subject to focused refinement with Relion and then were subject to local refinement with cryoSPARC, resulting in the 3D reconstruction of better quality on the interface between S-ECD and PD. 

The resolution was estimated with the gold-standard Fourier shell correlation 0.143 criterion [48] with high-resolution noise substitution [49]. Refer to Materials and Methods, Appendix A for details of data collection and processing.

### 2.4. Model Building and Structure Refinement

For model building of the complexes of BA.2/3/5-SA, the atomic model of the T-ACE-S(p) (PDB ID: 7CT5) was used as template, which was molecular dynamics flexible fitted [50] into the whole cryo-EM map of the complex and the focused-refined cryo-EM map of the RBD-PD sub-complex, respectively. Each residue was manually checked with the chemical properties taken into consideration during model building. Several segments, whose corresponding densities were invisible, were not modeled. Structural refinement was performed in Phenix [51] with secondary structure and geometry restraints to prevent overfitting. To monitor the potential overfitting, the model was refined against one of the two independent half maps from the gold-standard 3D refinement approach. Then, the refined model was tested against the other map. Statistics associated with data collection, 3D reconstruction, and model building are summarized in Appendix A.

## 3. Results

### 3.1. Biochemical Characterization and Structural Determination of the Complex Formation between Omicron Spike Proteins and ACE2-PD

To investigate the biochemical characteristics of Omicron subvariants, the binding affinity between their RBDs and ACE2-PD was compared. The monomeric human ACE2-PD binds to the RBDs of Omicron BA.1/2/5 with K_D_ of 11.5 ± 0.02 nM, 4.15 ± 0.01 nM, and 4.55 ± 0.02 nM, respectively, approximately two-to-four times higher than that of WT-RBD (18.40 ± 0.02 nM) [52]. Although the enhanced affinity with ACE2 by the Omicron variants may in part account for the increased transmissibility of Omicron variants, it cannot explain the particularly higher infectivity of BA.5 than BA.2. We then employed single-particle cryo-EM to solve the structures of the three trimeric S proteins each in complex with ACE2-PD.

We have previously solved the structure of the S protein in BA.1 bound with ACE2-PD at an overall resolution of 3.3 Å [42]. Identical protocols were used for protein expression, purification, and cryo-sample preparation for the three complexes. Please refer to Methods for details. The structures of the ACE2-PD-bound trimeric S proteins from BA.2, BA.3, and BA.5 were determined at overall resolutions of 3.3 Å, 3.4 Å, and 2.8 Å, respectively (Appendix A). For simplicity, we will refer to the three complexes as BA.1/2/3/5-SA.

The structures allow mapping of Omicron mutations, which are mainly distributed on the surface of the S protein (Figure 1B). Mutations on the spike glycoprotein of SARS-CoV-2 Omicron subvariants (BA.1/2/3/5) were mapped (Figure 1B). A total of 31 mutations from BA.1/2/3/5 can be revealed in our structures (Figure 1B). Among 20 mutations shared by four Omicron subvariants, 17 of them are clearly built. Five mutations (S371L, G496S, T547K, N856K, L981F) that only appeared in BA.1 are mapped. All mutations shared by BA.2/3/5 (S371F, D405N), mutations shared by BA.1/2/3 (Q493R), and all unique mutations from BA.5 (L452R, F486V) can be seen on the structure. As seven of the eight common mutations shared by BA.1/3 are located on flexible NTD domain, only the G446S located on the RBD was mapped. BA.2 and BA.5 have the highest number of shared mutations, and three of eight were mapped (A27S, T376A, R408S). The other mutations are invisible due to local flexibility, such as N679K and P681H near the furin cleavage site (residues 682–685), and A67V, H69del, and V70del on the loops in the NTD. 

### 3.2. Distinct Conformations of the S Proteins from the Four Subvariants

Despite identical sample preparation and data processing procedures, the four complexes exhibit distinct conformational preferences (Figure 2). In the 3D EM reconstruction for BA.1-SA, two RBDs are in the “up” conformation, while the third one is “down”. Consistent with previous studies, only the “up” RBDs are bound to ACE2-PD [9]. In contrast, all three RBDs are “up” in BA.2-SA and BA.5-SA. Therefore, three ACE2-PD molecules are bound to the trimeric BA.2/5 S protein. BA.3-SA represents a mixture, with 81.7% selected particles in all “up” states and the rest in the two “up” and one “down” conformation (Figure 2). 

Of note, previous structures of WT S protein in complexes with ACE2 show that WT-S tends to have either only one “up” and two “down” RBDs, or two “up” RBDs, among which only one is bound with ACE2-PD (67.7%) [13,21]. The increasing tendency of more “up” RBDs from WT to Omicron sublineages is consistent with the enhanced infectivity of the Omicron subvariants, particularly BA.2 and BA.5. As BA.3-SA represents a mixture of the BA.1-SA and BA.2/5-SA conformations, in what follows we will mainly focus on BA.2-SA and BA.5-SA for analysis.

### 3.3. Mutations of BA.2 and BA.5 Destabilize the “Down” Conformation of RBD

The molecular determinant for the “up” and “down” conformations of RBDs are critical to understanding the altered infectivities of the Omicron subvariants. We compared the structures of free S proteins that have all three RBDs in the “down” state from WT, BA.1, and BA.2 to figure out key mutations may affect the “up”/“down” conformation [53,54,55]. To facilitate structural illustration, the RBDs in the neighboring protomers will be referred to as RBD and RBD’ (Figure 3, top left). 

In the WT-S protein, two hydrogen bonds (H-bonds) are formed between Asp405 and Arg408 in RBD and Ser373 and the backbone carbonyl oxygen groups of Phe374 and Ser375 in RBD’ (Figure 3, top right). The mutations S373P and S375F in BA.1-S, which generate a more hydrophobic local environment, disrupt the interaction with Asp405 in the neighboring protomer. The carbonyl oxygen groups of Phe374 and Phe375 can still interact with the guanidinium group of Arg408 (Figure 3, bottom left). In BA.2-S, however, the mutation R408S disrupts the interaction with the backbone carbonyl groups. In addition, D405N is less favored by the nearby amide groups in the neighboring protomer (Figure 3, bottom right). The mutations, D405N and R408S of BA.4 are same as BA.2, which might explain the same three “up” RBD conformations of structure in BA.5.

These mutations together weaken the packing between the neighboring RBDs in the “down” conformation, thus promoting the tendency of “up” conformation of the S protein and suitable for ACE2 binding. The further-disrupted RBD-RBD’ interface in the “down” conformation thus affords a clue as to the all “up” conformation observed in our structure (Figure 2).

### 3.4. Several Omicron Mutations Strengthen the Interaction with ACE2

While the various conformational preferences of the trimeric S protein may account for the increased infectivity of the Omicron subvariant, especially BA.2 and BA.5, they cannot explain the higher affinity between the RBDs of all four Omicron subvariants and ACE2-PD, as the measurement was performed with isolated monomeric domain [52]. Therefore, we meticulously examined the mutations on the interface with ACE2. 

Structural analysis has identified 17 RBD residues on the interface with ACE2 [20,56,57]. Sequence alignment of the RBD from WT, Delta, and Omicron subvariants reveals that nine residues, Tyr449, Tyr453, Leu455, Phe456, Asn487, Tyr489, Tyr495, Thr500, and G502, remain identical in all the subvariants, while six mutated in BA.1, BA.2, and BA.3 subvariants, including K417N, S477N, Q493R, Q498R, N501Y, and Y505H. BA.1 has an additional mutation, G496S, while BA.5 has an additional mutation, F486V. Both BA.1 and BA.5 retain Gln493, which is the same as the WT S protein. These mutations have the potential to impact the interaction network between the RBD and ACE2 (Figure 4 and Appendix A).

Compared with the WT S protein, the mutations S477N, Q493R, and Q498R in the Omicron RBD introduce new polar interactions with Ser19, Glu35, and Asp38 in ACE2, respectively. These additional contacts, particularly the strong salt bridges mediated by Arg493 and Arg498, may not only compensate for the lost interactions of Lys417, Asn501, and Tyr505 in the WT S protein, respectively, with Asp30, Tyr41, and Arg393 in ACE2, but result in a two-to-fourfold net increase in affinity (Figure 4B–D). Furthermore, the BA.5 mutation F486V maintains the Van der Waals forces interaction with ACE2, and Gln493 is capable of forming a hydrogen bond with His34 of ACE2. However, this interaction may be influenced by the local environment due to the L452R mutation in BA.5 (Figure 4B,C). 

### 3.5. The Immune Evasion Mechanism of S309 and S2K146

Next, we examined the resolved mutations to look for clues to the immune evasion mechanism of Omicron subvariants. Many of the surface-mapping mutations (Figure 1B) have been analyzed previously [13,58]. To avoid redundancy, here we mainly focus on the distinct responses to antibody S309 and S2K146. The antibody S309 can still neutralize BA.1, but not the BA.2 and BA.5 subvariants, and antibody S2K146 could still neutralize BA.1, BA.2, and BA.3, but not BA.5 [26,30,31]. 

The authorized monoclonal antibody S309 has been shown to effectively neutralize several SARS-CoV-2 variants, including BA.1. However, its neutralizing activity against BA.2 and BA.5 is largely compromised [23]. In contrast, another broadly neutralizing antibody, LY-CoV1404, is still potent in neutralizing all Omicron sublineages [23,31,32]. Both S309 and LY-CoV1404 belong to Class 3 that targets the outside ACE2 binding site and recognize both “up” and “down” RBDs. 

Among all mutations on the RBD, only G339D is positioned in the epitope of S309. However, it is shared by BA.1, BA.2, and BA.5 (Figure 5A and Appendix A). As a result, this mutation does not account for the different sensitivities of BA.1, BA.2, and BA.5 to S309. We therefore compared the maps for BA.1-SA, BA.2-SA, and BA.5-SA for potential allosteric mutation sites. We observed an evident shift of the glycosylation moieties linked to Asn343 among these three maps. Furthermore, the conformation of this glycan is nearly identical between BA.1 and WT (Figure 5B and Appendix A). This difference can be potentially important as the Asn343 glycan is directly involved in S309 recognition (Figure 5A) [59]. 

Scrutiny of the local environment suggests that the shift of the N343-glycan in BA.2-SA and BA.5-SA is a direct consequence of the mutation S371F, which is found in BA.2, BA.3, and BA.5, but not BA.1. The bulky hydrophobic ring of Phe371 presents a steric hindrance to the polar sugar moieties nearby. Therefore, the N343-glycan has to move away. Notably, this shift would result in a clash with the antibody S309, making the binding less favored (Figure 5C and Appendix A). Our comparative structural study thus reveals an allosteric mechanism of the escape from S309 by BA.2, BA.3, and BA.5 through the single point mutation S371F, which is not directly positioned on the S309 epitope.

Antibody S2K146 can neutralize BA.2 and BA.3, but significantly loses its neutralizing activity against BA.5. The compromised neutralization of S2K146 to BA.5 might be directly affected by the mutations, L452R and F486V of BA.5 (Figure 5D). When aligned the epitopes of S2K146 with WT, BA.2, and BA.5, the F486V obviously disrupts the pie-pie interaction net that involves Phe486 of the RBD, Trp92 of the light chain, and Trp105 and Tyr106 of the heavy chain (Figure 5E, left). The L452R could cause a clash with Arg102 of the heavy chain (Figure 5E, right).

LY-CoV1404 preserves neutralizing activity for all five Omicron subvariants. It is noted that four Omicron mutations, N440K, G446S, Q498R, and N501Y, are mapped to the epitope of LY-CoV1404 (Appendix A). Instead of disrupting interactions with the antibody, Lys440 and Arg498 of BA.2 form H-bonds with Tyr35 and Thr96 of LY-CoV1404, respectively. The mutation N501Y does not affect the interaction as neither Asn nor Tyr directly participates in the interaction with the antibody. In addition, G446S, a common mutation in BA.1 and BA.3, might lead to the interaction between Ser446 and Arg60 of the heavy chain in LY-CoV1404 (Appendix A). As seen, none of these mutations impair the interaction with the antibody, explaining the broad neutralizing mechanism of LY-CoV1404.

## 4. Discussion

In this study, we attempt to provide clues to two critical questions regarding the Omicron variants through comparative structural analysis of ACE2-bound S proteins from BA.1/2/3/5, First, we examined the molecular basis for the increased transmissibility of Omicron subvariants, particularly BA.2 and BA.5. Second, we examined the mechanism for distinct responses to antibody neutralization. 

In addition to the varied affinity with the cellular receptor ACE2, the switch to control the “up” and “down” states of RBD is critical for the infectivity of SARS-CoV-2. Our structural investigation reveals that in the presence of the PD of ACE2, RBDs in all three protomers of BA.2/5 S proteins exhibit the “up” conformation, whereas only two are “up” in BA.1-S. This observation may partially account for the increased infectivity of BA.2 and BA.5. Previous studies have shown that cleavage of the furin cleavage site promotes the “up” conformation of RBD [21,60]. Here, we discover that a preference for the “up” state of RBDs bound with ACE2 may be achieved through destabilizing the “down” state packing and facilitating ACE2 binding. Therefore, the Omicron subvariants may acquire enhanced infectivity through both tighter binding to ACE2 and an increased tendency for the “up” state. Detailed analysis of the disruptive mutations on the RBD interface in their “down” state is consistent with their differential preference for the “up” state.

The COVID-19 pandemic has continued to spread across the world for over three years [61]. Under the selective pressure of the immune system, variants of SARS-CoV-2 have evolved to evade the antibody immunity elicited by the vaccine immunization or natural viral infection [62,63,64,65,66]. A large number of mutations of BA.1, BA.2, BA.3, and BA.5 are distributed on the hotspots of epitopes, providing a mechanistic basis for the immune evasion from many therapeutic antibodies and vaccinations (Figure 1B). Out of expectation, an allosteric mechanism is discovered to allow BA.2, BA.3, and BA.5 to escape the neutralization of S309 through the mutation of S371F, which alters the conformation of a nearby glycan to hinder antibody binding. Recently, a molecular dynamics simulation study also suggested that BA.1 mutations can increase the interactions of S309 with glycan but BA.2 or BA.4/5 mutations can decrease the binding, highlighting the role of glycosylation in antibody recognition [67]. As well as this, the additional mutations of BA.5 further reduce the neutralization activity of S309, suggesting an unexpected fitness of SARS-CoV-2.

During the revision of this manuscript, several research groups have reported structures of the BA.2 and BA.5 subvariants [31,52,68,69]. Saville et al. presented the structures of the S proteins in both BA.1 and BA.2, as well as the BA.2 S protein in complex with the PD of mouse ACE2 [69]. Xu et al. reported the BA.2 S protein in complex with the PD of human ACE2, revealing two distinct structural states: three PDs bound to three “up” RBDs and two PDs bound to two “up” RBDs. They also investigated the binding of BA.2 and BA.1 S proteins with the PD of mouse ACE2 [68]. These findings suggest that the Omicron variants may have originated from mouse hosts. Additionally, Cao et al. also reported the S protein structures of BA.2.12.1, BA.4, and BA.5 and explored their potential for immune evasion, particularly emphasizing the importance of the S371F mutation in BA.2/4/5 in evading immune response [31]. It is worth noting that the different binding patterns of BA.2 observed in the studies by Xu et al. and ours could be attributed to differences in protein purification strategies and incubation conditions. This highlights the significance of systematically examining these structures. Comparing results obtained under consistent experimental conditions can lead to more robust conclusions.

Based on the obtained structures of the spike protein, we have observed that different subvariants of Omicron exhibit a preference for distinct conformations. Specifically, under the same experimental conditions, recent subvariants BA.2 and BA.5 have shown a higher proportion of RBDs in the “up” state, which is capable of binding to the receptor. This consistency aligns with the increased infectivity observed in BA.2 and BA.5 compared to BA.1, suggesting that mutations in the S protein promote a higher tendency for RBDs to adopt the “up” conformation. This, in turn, increases the likelihood of receptor binding and contributes to the enhanced transmissibility of the Omicron variant. However, it is important to note that in their natural state, the “up” and “down” conformations of RBDs are expected to dynamically interconvert. We benefitted from cryo-electron microscopy, which enabled us to observe and statistically analyze the proportions of different conformations at the moment of freezing. However, the lack of suitable observational and experimental techniques make it challenging to study the natural conformations, poses, and propensities of RBDs. Consequently, further molecular-level validation of the impact of different mutations on conformation and viral infectivity remains elusive. In conclusion, our studies provide insights into the molecular basis of S protein recognition by the ACE2 receptor in Omicron subvariants.

## Figures and Tables

**Figure 1 viruses-15-01398-f001:**
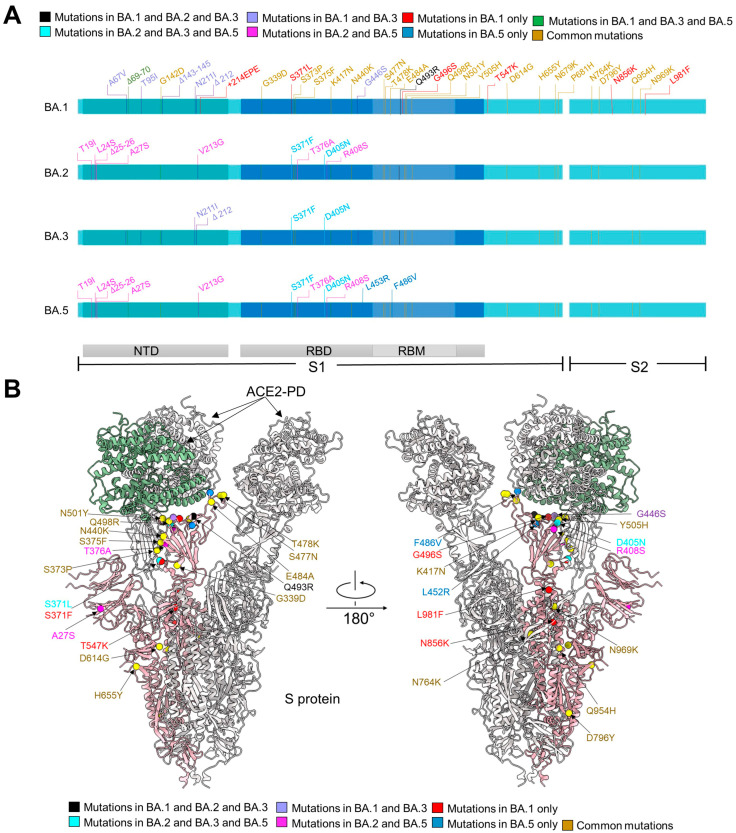
The distribution of mutations of S protein from Omicron sublineages. (**A**) Summary of mutations mapped to the extracellular domain of the spike protein (S-ECD) from BA.1, and different mutations from BA.2, BA.3, and BA.5 with respect to BA.1. They share 20 common mutations, which are labeled brown. Mutations shared by BA.1/2/3, BA.2/3/5, or BA.1/3/5 are colored black, cyan, or green, respectively. The 8 mutations in BA.1 and BA.3, but not BA.2 and BA.5, are colored purple and the mutations only shared by BA.2 and BA.5 are labeled magenta. The unique mutations in BA.1 and BA.5 are colored red and blue, respectively. NTD: amino terminal domain. RBD: receptor-binding domain. RBM: receptor-binding motif. (**B**) Mapping of mutations to the S protein of Omicron subvariants. Mutations color strategies are same as these in (**A**).

**Figure 2 viruses-15-01398-f002:**
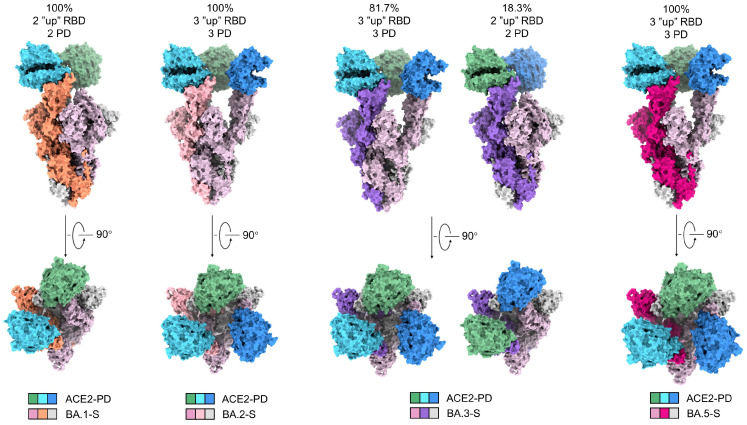
The trimeric S proteins from the four Omicron subvariants display distinct conformations and stoichiometric ratios with ACE2-PD. Shown here is surface presentation of domain-colored cryo-EM structures of S-ECD from Omicron BA.1, BA.2, BA.3, and BA.5, respectively, in complex with the PD of ACE2. The three RBDs in BA.2-S and BA.5-S are all “up” and associate with three ACE2-PD. Two RBDs are “up” and one “down” in BA.1-S, and only the two “up” RBDs are associated with ACE2-PD. The majority (81.7%) of BA.3-S are identical with BA.2/5-S and the rest are the same as BA.1-S.

**Figure 3 viruses-15-01398-f003:**
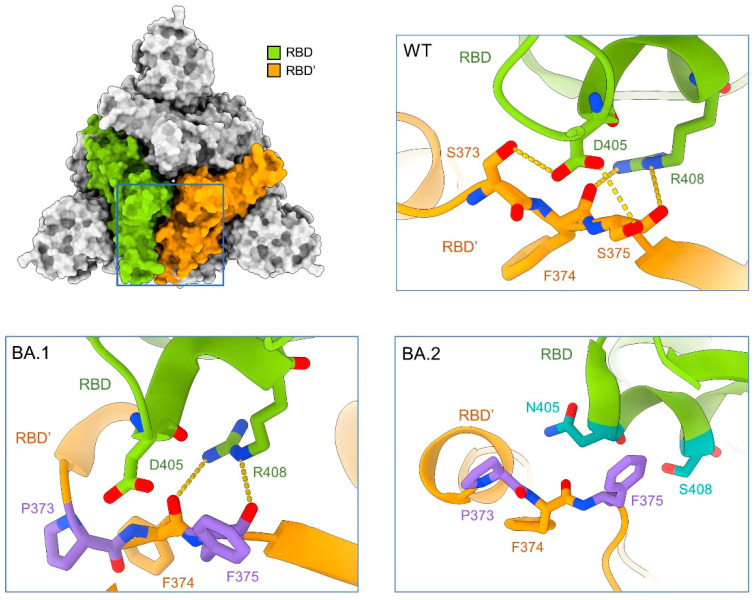
Mutations on the RBD-RBD’ interface in the Omicron subvariants may destabilize the “down” conformation, thus contributing to their enhanced infectivity by favoring the “up” conformation. Structure of the top view of the “all down” S-ECD from the original SARS-CoV-2 strain (designated as WT, PDB: 6ZB5) is shown on the top left. *Insets*: Enlarged views of the interface between the adjacent RBDs, labelled as RBD and RBD’, from WT (top right), BA.1 (bottom left, PDB: 7TF8), and BA.2 (bottom right, PDB: 7UB0). Residues on the interface are shown as sticks, green (RBD), or orange (RBD’) for invariant ones and dark green (RBD) or purple (RBD’) for mutated ones.

**Figure 4 viruses-15-01398-f004:**
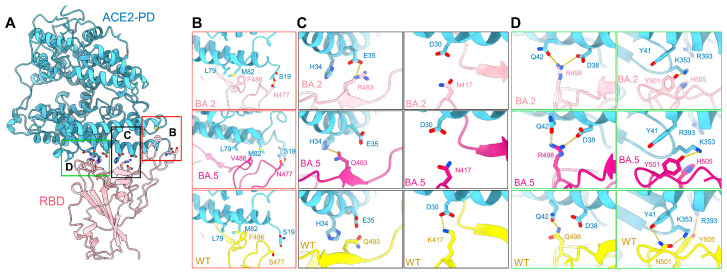
Mutations on the interface of ACE2 and BA.2/5-S underlie their enhanced affinity. (**A**) Point mutations of BA.2-S strengthen its interaction with ACE2. Detailed comparison of the ACE2-interacting residues between the BA.2, BA.5, and WT S proteins are shown in panels (**B**–**D**). (**B**) S477N of BA.2/5-S could form an additional hydrogen bond with Ser19 of ACE2. F486V keeps the Van der Waals forces with Met82 and Leu79 of ACE2. (**C**) Q493R of BA.2-S results in a salt bridge with ACE2-Glu35 and Gln493 of BA.5 interacts with ACE2-His34, although K417N weakens the original interaction with ACE2-Asp30. (**D**) Q498R of BA.2/5 could form a salt bridge with ACE2-Asp38, and N501Y leads to a rearrangement of the local interactions with Tyr41 and Ly353 in ACE2. Y505H breaks the hydrogen bond with R393 in ACE2. The PDB ID for the WT structure is 6M17.

**Figure 5 viruses-15-01398-f005:**
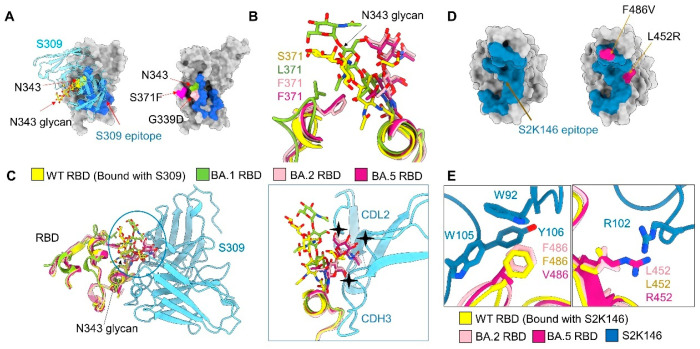
Potential mechanism for S309 and S2K146 escaping neutralization. (**A**) There is no BA.2/5-unique mutation on the S309 epitope. *Left*: A glycan on Asn343, shown as yellow sticks, is recognized by antibody S309. Shown here is the structure of WT RBD bound to antibody S309 (PDB code: 6WPT). The S309 epitope is colored blue. *Right*: Omicron subvariant mutations in the vicinity of the S309 epitope. G339D, which is common to BA.1, BA.2, and BA.5, is the only one mapped to the S309 epitope. S371F is a mutation in BA.2, BA.3, and BA.5, but missing in BA.1. (**B**) The glycosylation moieties of Asn343 of RBD is pushed away due to the steric hindrance as a result of S371F mutation in BA.2/5-S. (**C**) Mutation S371F may underlie BA.2, BA.3, and BA.5′s evasion from S309 neutralization by altering the conformation of Asn343 glycan. Structures of the RBD from the WT in complex with antibodies S309, BA.1, BA.2, and BA.5 are superimposed (left). The Asn343 glycan in BA.1 exhibits a similar conformation as that in WT, while that in BA.2 and BA.5 has to move away due to the potential steric clash with Phe371. The shifted Asn343 glycan in BA.2 and BA.5 would clash with antibody S309 (right), hence impeding its recognition of the epitope. (**D**) There are two unique mutations of BA.5 on the S2K146 epitope. Left: The S2K146 epitope is colored steel blue (PDB code: 7TAS). Right: unique mutations of BA.5 in the vicinity of the S2K146 epitope. (**E**) F486V in BA.5 disrupt the pie–pie interaction net that involves Phe486 of RBD, Trp92 of light chain, Trp105 and Tyr106 of heavy chain and the L452R could cause a clash with Arg102 of heavy chain.

## Data Availability

Atomic coordinates and cryo-EM density maps of Omicron BA.2 S protein in complex with PD of ACE2 (PDB: 7Y1Y, whole map: EMD-33575, map focused on the interface between BA.2 RBD and ACE2: EMD-33579). Two “up” RBD state of BA.3 S protein in complex with PD of ACE2 (PDB: 7Y20, whole map: EMD-33577) and three “up” RBD state of BA.3 S protein in complex with PD of ACE2 (PDB: 7Y1Z, whole map: EMD-33576, map focused on the interface between BA.3 RBD and ACE2: EMD-33580) have been deposited to the Protein Data Bank (http://www.rcsb.org) and the Electron Microscopy Data Bank (https://www.ebi.ac.uk/pdbe/emdb/), respectively. BA.5 S protein in complex with PD of ACE2 (PDB: 7Y21, whole map: EMD-33578, map focused on the interface between BA.5 RBD and ACE2: EMD-33581). The other PDB and EMDB IDs can be found in Appendix A.

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
