# Peer review of "Structural Basis for the Enhanced Infectivity and Immune Evasion of Omicron Subvariants"

_viruses, 2023, doi:10.3390/v15061398_

Round 1

Reviewer 1 Report

Yaning Li et al. have reported the cryo-EM structures of the interface complexes of the trimeric S proteins and ACE2 receptor for Omicron subvariants BA.1, BA.2, BA.3, and BA.4/BA.5. Additionally, they have analyzed the epitopes for certain mAbs to investigate the immune evasion mechanism of Omicron subvariants. The methods are well designed and clearly described, the results are well presented, and the discussion is based on the results, but it is required more related studies to enhance the conclusion further. This study is very interesting and important for providing the molecular basis for the increased transmissibility and immune evasion of Omicron mutations. However, this MS should be major revised to address the following comments:

Major comments:

1) In the abstract, the sentences in lines 14 and 15 should be rephrased. For example, use “Their S proteins have more than 30 mutations compared to the original WT strain.” for the second sentence.

Also, the authors should provide at least one more sentence about their findings, particularly the molecular basis of increased infectivity and how the Omicron mutations at the RBD Omicron variants preferred to bind with ACE2 or escape for mAbs.

2) Figure 1(A) should be improved to make it easier to understand using the following suggestions:

i) Using the same color for all Omicron variants with different colors of each S-protein domain. Let’s say, using the same color for BA.1 in BA.2, BA.3, and BA.5.

ii) Instead of using arrows to show mutations, use the lines.

iii) Do not label the common mutations in BA.2, BA.3, and BA.5 that already belong to BA.1 and only show the lines near to their locations. Or follow the same style that is used in this website https://covdb.stanford.edu/variants/omicron_ba_1_3/ by showing the only different mutations with respect to BA.1.

-In addition, the colors of the legends in Figures 1(A) and (B) differ. It is better if they are identical. Here, the mutations in Figure 1(A) should follow the same color as in Figure 1(B). For instance, use yellow color for common mutations in both figures and so on. Following these changes, the figure caption should be modified.

3) In line 90, the authors should also mention that several BA.2, BA.3, and BA.5 structures have been published, and rephrase this sentence “However, these structures cannot explain the enhanced infectivity and acquired antibody resistance of BA.2, BA.3, and BA.5”. You should focus on the main aim of this work which is “to provide the molecular basis for the increased transmissibility and immune evasion of Omicron mutations”.

-Also, the authors should show the new findings that other BA.2, BA.3, and BA.5 structures do not report, especially in the discussion section. For example, see and cite the following references:

https://doi.org/10.1016/j.celrep.2022.111964.

https://doi.org/10.1038/s41422-022-00672-4.

https://doi.org/10.1038/s41586-022-04980-y

-Could the authors also discuss the differences and similarities in the "up" and "down" conformations of Omicron subvariants observed in their study and in the references above?

4) The references in the materials and methods section are missing from the manuscript, and they are arranged incorrectly ("see lines 154 to 185"). Please correct this.

5) In line 280, the authors stated that “Structural analysis has identified 9 RBD residues on the interface with ACE2.” Did the authors compare this result to previous studies? To my knowledge, other studies determined more than this number of “9 residues”. Please check out/compare with the following references:

https://doi.org/10.1038/s41586-020-2180-5

https://doi.org/10.1021/acs.jcim.1c00560

-In line 323, the authors observed “an evident shift of the glycosylation moieties linked to Asn343 among these three maps.”, this observation requires more evidence, which can be found in the following preprint:

https://doi.org/10.1101/2022.12.25.521903

6)  Limitations of the work should be mentioned in the discussion section.

Minor comments:

1)      Change the following:

-In line 16, change “BA.1-BA.5” to “BA.1, BA.2, BA.3 and BA.4/BA.5”.

-In line 34, change “BA.2” to “BA.1”.

-In lines 49-52, the sentences should be modified to include a new update of Omicron subvariants.

-Replace "sub-lineage(s)" with "sublineage(s)" across the manuscript.

-Please avoid using long sentences like the one in lines 81-85, thus split this sentence into two by using a full dot after "refs.12,23." with appropriate modifications.

-In line 103, a.a should be defined as an amino acid.

-In line 131, add “and” between “HEPES (pH 7.0)” and “150 mM NaCl.” Do the same for the one in lines 138, 141, and 145.

-In line 180, change “templates” to “a template”.

-The sentence on lines 116-117 is unclear because "G446S" is part of RBD, not NTD; please modify this sentence to avoid confusion.

-Figure captions 4 and 5 are in bold, while the others are not. Please stick to one style "bold caption".

-In line 300, change “Waals” to “Van der Waals”.

-In line 337, change “Fig.5d” to “Figure 5(D)”
-In line 418, change “his” to “This”.

-In lines 470-471, check ref.#25

2)      In line 24, the authors should mention the keywords.

3)      There is inconsistency in the citation style. The citation is appeared with “ref.XX” and “XX”, please use only “XX” style. Correct the citation in lines 33,52,83,85,92,158, and 164.

4)      These sentences should be well references:

-In line 106, “A “GSAS” mutation at residues 682 to 685 was 106 introduced into ECD to prevent the host furin protease digestion.”

-In line 115, “The mutants were generated with a standard two-step PCR-based strategy.”

-In line 116, “All the plasmids used to transfect cells were prepared by GoldHi EndoFree Plasmid Maxi Kit (CWBIO).”

5)      In Reference, the authors should update all the preprints that have recently been published.

6)      In the supplementary file on page 1, please add the manuscript title and name of authors with their affiliations.

The authors should carefully check for typos and avoid using long sentences. 

Author Response

Referee #1

Major comments:

1) In the abstract, the sentences in lines 14 and 15 should be rephrased. For example, use “Their S proteins have more than 30 mutations compared to the original WT strain.” for the second sentence.

Also, the authors should provide at least one more sentence about their findings, particularly the molecular basis of increased infectivity and how the Omicron mutations at the RBD Omicron variants preferred to bind with ACE2 or escape for mAbs.

Point taken. We have rephrased these sentences and changed the summary about our findings as “The Omicron variants of SARS-CoV-2 have emerged as the dominant strains worldwide, causing the COVID-19 pandemic. Each Omicron subvariant contains at least 30 mutations on the Spike protein (S protein) compared to the original wild-type (WT) strain. and “By analyzing the position of the glycan modification on Asn343, which is located at the S309 epitopes, we have uncovered the underlying immune evasion mechanism of the Omicron subvariants. Our findings provide a molecular basis of high infectivity and immune evasion of Omicron subvariants, thereby offering insights into potential therapeutic interventions against SARS-CoV-2 variants.

2) Figure 1(A) should be improved to make it easier to understand using the following suggestions:

  1. i) Using the same color for all Omicron variants with different colors of each S-protein domain. Let’s say, using the same color for BA.1 in BA.2, BA.3, and BA.5.
  2. ii) Instead of using arrows to show mutations, use the lines.

iii) Do not label the common mutations in BA.2, BA.3, and BA.5 that already belong to BA.1 and only show the lines near to their locations. Or follow the same style that is used in this website https://covdb.stanford.edu/variants/omicron_ba_1_3/ by showing the only different mutations with respect to BA.1. 

-In addition, the colors of the legends in Figures 1(A) and (B) differ. It is better if they are identical. Here, the mutations in Figure 1(A) should follow the same color as in Figure 1(B). For instance, use yellow color for common mutations in both figures and so on. Following these changes, the figure caption should be modified.

Point taken. We have made changes to the Figure 1(A) according to all above suggestions.

3) In line 90, the authors should also mention that several BA.2, BA.3, and BA.5 structures have been published, and rephrase this sentence “However, these structures cannot explain the enhanced infectivity and acquired antibody resistance of BA.2, BA.3, and BA.5”. You should focus on the main aim of this work which is “to provide the molecular basis for the increased transmissibility and immune evasion of Omicron mutations”.   

Point taken. We have rephrased this sentence as “additional structures involving BA.2, BA.3, and BA.5 in complex with ACE2 are necessary to establish the molecular foundation for the heightened transmissibility and immune evasion observed in Omicron mutations.

-Also, the authors should show the new findings that other BA.2, BA.3, and BA.5 structures do not report, especially in the discussion section. For example, see and cite the following references:    

https://doi.org/10.1016/j.celrep.2022.111964.

https://doi.org/10.1038/s41422-022-00672-4.

https://doi.org/10.1038/s41586-022-04980-y

-Could the authors also discuss the differences and similarities in the "up" and "down" conformations of Omicron subvariants observed in their study and in the references above?

We thank this reviewer for this insightful suggestion. We have added the discussion as “During the revision of this manuscript, several research groups have reported structures of the BA.2 and BA.5 subvariants. Saville et al. presented the structures of the S protein in both BA.1 and BA.2, as well as the BA.2 S protein in complex with the PD of mouse ACE2. Xu et al. reported the BA.2 S protein in complex with the PD of human ACE2, revealing two distinct structural states: three PDs bound to three "up" RBDs and two PDs bound to two "up" RBDs. They also investigated the binding of BA.2 and BA.1 S proteins with the PD of mouse ACE2. These findings suggest that the Omicron variants may have originated from mouse hosts. Additionally, Cao et al. also reported the S protein structures of BA.2.12.1, BA.4, and BA.5 and explored their potential for immune evasion, particularly emphasizing the importance of the S371F mutation in BA.2/4/5 in evading immune response. It is worth noting that the different binding patterns of BA.2 observed in the studies by Xu et al. and ours could be attributed to differences in protein purification strategies and incubation conditions. This highlights the significance of systematically examining these structures. Comparing results obtained under consistent experimental conditions can lead to more robust conclusions.

4) The references in the materials and methods section are missing from the manuscript, and they are arranged incorrectly ("see lines 154 to 185"). Please correct this.

Point taken. We have corrected this at the revised manuscript.

5) In line 280, the authors stated that “Structural analysis has identified 9 RBD residues on the interface with ACE2.” Did the authors compare this result to previous studies? To my knowledge, other studies determined more than this number of “9 residues”. Please check out/compare with the following references:

https://doi.org/10.1038/s41586-020-2180-5

https://doi.org/10.1021/acs.jcim.1c00560

Point taken. We have compared with these two references and added the results into this manuscript. Structural analysis has identified 17 RBD residues on the interface with ACE2. Sequence alignment of the RBD from WT, Delta, and Omicron subvariants reveals that 9 residues, Tyr449, Tyr453, Leu455, Phe456, Asn487, Tyr489, Tyr495, Thr500 and G502, remain identical in all the subvariants, while 6 mutated in BA.1, BA.2, and BA.3 subvariants, including K417N, S477N, Q493R, Q498R, N501Y and Y505H. BA.1 has an additional mutation, G496S, while BA.5 has an additional mutation, F486V. Both BA.1 and BA.5 retain Gln493, which is the same as the WT S protein. These mutations have the potential to impact the interaction network between the RBD and ACE2 (Figures 4 and S7).

Comparing with WT S protein, the mutations S477N, Q493R, and Q498R in the Omicron RBD introduce new polar interactions with Ser19, Glu35, and Asp38 in ACE2, respectively. These additional contacts, particularly the strong salt bridges mediated by Arg493 and Arg498, may not only compensate for the lost interactions of Lys417, Asn501 and Tyr505 in the WT S protein respectively with Asp30, Tyr41 and Arg393 in ACE2, but result in a 2-4 fold net increase in affinity (Figures 4B-D). Furthermore, the BA.5 mutation F486V maintains the Van der Waals forces interaction with ACE2, and Gln493 is capable of forming a hydrogen bond with His34 of ACE2. However, this interaction may be influenced by the local environment due to the L452R mutation in BA.5 (Figures 4B, 4C).

-In line 323, the authors observed “an evident shift of the glycosylation moieties linked to Asn343 among these three maps.”, this observation requires more evidence, which can be found in the following preprint:

https://doi.org/10.1101/2022.12.25.521903

We thank this reviewer for this suggestion. We have added the statement in discussion as “Recently, a molecular dynamics simulation study also suggested that BA.1 mutations can increase the interactions of S309 with glycan but BA.2 or BA.4/5 mutations can decrease the binding, highlighting the role of glycosylation in antibody recognition.

6)  Limitations of the work should be mentioned in the discussion section.  

We thank this reviewer for this suggestion. We have added the limitations of the work at the final paragraph of discussion as“Based on the obtained structures of the spike protein, we have observed that different subvariants of Omicron exhibit a preference for distinct conformations. Specifically, under the same experimental conditions, recent subvariants BA.2 and BA.5 have shown a higher proportion of RBDs in the "up" state, which is capable of binding to the receptor. This con-sistency aligns with the increased infectivity observed in BA.2 and BA.5 compared to BA.1, suggesting that mutations in the S protein promote a higher tendency for RBDs to adopt the "up" conformation. This, in turn, increases the likelihood of receptor binding and con-tributes to the enhanced transmissibility of the Omicron variant. However, it is important to note that in their natural state, the up and down conformations of RBDs are expected to dynamically interconvert. Benefit from cryo-electron microscopywe enables us to observe and statistically analyze the proportions of different conformations at the moment of freezing. But the lack of suitable observational and experimental techniques make it challenging to studying the natural conformations poses propensity of RBDs to adopt different conformations poses challenges due to the lack of suitable observational and experimental techniques. Consequently, further molecular-level validation of the impact of different mutations on conformation and viral infectivity remains elusive. In conclusion, our studies provide insights into the molecular basis of S protein recognition by the ACE2 receptor in Omicron subvariants.

Minor comments:

1)      Change the following:

-In line 16, change “BA.1-BA.5” to “BA.1, BA.2, BA.3 and BA.4/BA.5”.

Point taken. We have changed “BA.1-BA.5” to “BA.1, BA.2, BA.3 and BA.4/BA.5”.

-In line 34, change “BA.2” to “BA.1”.

Point taken. We have changed this “BA.2”.

-In lines 49-52, the sentences should be modified to include a new update of Omicron subvariants.

Point taken. We have included new update subvariants XBB. “Recently, a newly emerged sublineage, Omicron XBB, characterized by high antibody evasion abilities, has gained dominance in the population.

-Replace "sub-lineage(s)" with "sublineage(s)" across the manuscript.

Point taken. We have changed all “sub-lineage(s)” to “sublineage(s)”

-Please avoid using long sentences like the one in lines 81-85, thus split this sentence into two by using a full dot after "refs.12,23." with appropriate modifications.

Point taken. We have split this sentence into two after "refs.12,23."

-In line 103, a.a should be defined as an amino acid.

Point taken. We have changed all “a.a” to “amino acid”.

-In line 131, add “and” between “HEPES (pH 7.0)” and “150 mM NaCl.” Do the same for the one in lines 138, 141, and 145.

Point taken. We have added “and” between “HEPES (pH 7.0)” and “150 mM NaCl.”

-In line 180, change “templates” to “a template”.

Point taken. We have changed “templates” to “a template”

-The sentence on lines 116-117 is unclear because "G446S" is part of RBD, not NTD; please modify this sentence to avoid confusion.

Point taken. We have changed the sentence to “As seven of the eight common mutations shared by BA.1/3 are located on flexible NTD domain, only the G446S located on RBD was mapped.

-Figure captions 4 and 5 are in bold, while the others are not. Please stick to one style "bold caption".

Point taken. We have styled Figure captions 1,2 and 3 as “bold”.

-In line 300, change “Waals” to “Van der Waals”.

Point taken. We have change “Waals” to “Van der Waals”.

-In line 337, change “Fig.5d” to “Figure 5(D)”

Point taken. We have changed “Fig.5d” to “Figure 5D”.

-In line 418, change “his” to “This”.

Point taken. We have changed “his” to “This”.

-In lines 470-471, check ref.#25

Point taken. We have checked and corrected it.

2)      In line 24, the authors should mention the keywords.

Point taken. We give five keywords: “SARS-CoV-2; Omicron subvariant; Spike; ACE2; Immune evasion”

3)      There is inconsistency in the citation style. The citation is appeared with “ref.XX” and “XX”, please use only “XX” style. Correct the citation in lines 33,52,83,85,92,158, and 164.  ---LYN

Point taken. We have corrected them.

4)      These sentences should be well references:

-In line 106, “A “GSAS” mutation at residues 682 to 685 was 106 introduced into ECD to prevent the host furin protease digestion.”

-In line 115, “The mutants were generated with a standard two-step PCR-based strategy.”

-In line 116, “All the plasmids used to transfect cells were prepared by GoldHi EndoFree Plasmid Maxi Kit (CWBIO).”

Point taken. We have added related references in the revised manuscript.

5)      In Reference, the authors should update all the preprints that have recently been published.

Point taken. We have updated reference.

6)      In the supplementary file on page 1, please add the manuscript title and name of authors with their affiliations.

Point taken. We have added title and authors in the revised supplementary file.

Reviewer 2 Report

The authors in this study provide structural data and information about the S protein of SARS-CoV-2 omicron subvariants.

The Cryo-EM data is very insightful with info about the up and down conformations of S protein. However, there are certain questions which must be addressed:

-Further experiments, other than Cryeo-EM, must be provided to directly correlate the up conformation with increased heterogeneity. The example of such exeriments would include mutagenesis to chenge the conformation and testing for immunogenicity etc.

-In the absence of further orthogonal methods and experiments, this would only be a suggestion in the direction of providing a molecular basis.

The English language is generally fine however it would be appreciated if some paragraphs are continuing the discussion or connect with previous paragraphs, within a defined context.

Author Response

Referee #2

Major comments:

The Cryo-EM data is very insightful with info about the up and down conformations of S protein. However, there are certain questions which must be addressed:

-Further experiments, other than Cryeo-EM, must be provided to directly correlate the up conformation with increased heterogeneity. The example of such exeriments would include mutagenesis to chenge the conformation and testing for immunogenicity etc.??

-In the absence of further orthogonal methods and experiments, this would only be a suggestion in the direction of providing a molecular basis.

We thank this reviewer for this good suggestion. The “up” and “down” RBD state of S protein is an important factor for investigating the infectivity. The more “up” RBD means the increasing the probability of receptor binding and contributing to the enhanced transmissibility of Omicron, which could be revealed by the structural analysis. The experiment suggested by this reviewer could obviously help understand the infection process of Omicron variants, but unfortunately, we couldn’t perform the virus infection assay in our lab until now. So we have tune down our finding of this manuscript as “Our findings provide the molecular basis of high infectivity and immune evasion of Omicron subvariants.

Round 2

Reviewer 1 Report

The authors have satisfactorily addressed all of my concerns, and the revised version of the manuscript is improved significantly. Therefore, I would recommend the manuscript for publication.

The English language is fine, although there are a few typos that should be double-checked.